# COVID-19 is linked to changes in the time–space dimension of human mobility

Clodomir Santana [1], Federico Botta [1,2], Hugo Barbosa [1], Filippo Privitera[3], Ronaldo Menezes [1,2,4] & Riccardo Di Clemente [1,2,5] ✉

Socio-economic constructs and urban topology are crucial drivers of human mobility patterns. During the coronavirus disease 2019 pandemic, these patterns were reshaped in their components: the spatial dimension represented by the daily travelled distance, and the temporal dimension expressed as the synchronization time of commuting routines. Here, leveraging location-based data from de-identified mobile phone users, we observed that, during lockdowns restrictions, the decrease of spatial mobility is interwoven with the emergence of asynchronous mobility dynamics. The lifting of restriction in urban mobility allowed a faster recovery of the spatial dimension compared with the temporal one. Moreover, the recovery in mobility was different depending on urbanization levels and economic stratification. In rural and low-income areas, the spatial mobility dimension suffered a more considerable disruption when compared with urbanized and high-income areas. In contrast, the temporal dimension was more affected in urbanized and high-income areas than in rural and low-income areas.

The places we visit[1–3], the products we purchase[4–6] and the people we interact with[7–9], among other activities, produce digital records of our daily activities. Once decoded and analysed, this digital fingerprint provides a new ground to portray urban dynamics[10,11]. In particular, the sequence of locations gathered from mobile phone devices via call detail records (CDR) and location-based service (LBS) data offers a unique opportunity to assess a broad time span of people's urban activities in almost real time, overcoming the limitations of surveys and censuses[12,13]. CDR and LBS give us information about people's daily motifs across urban locations[14], their attitude in exploring different places[15], the route of their commutes[16,17] and the purpose of their urban journey[18]. The location of people in cities is predictable[19] and is strictly connected with the circadian rhythms of social activities[20], as well as home and work locations. The spatio-temporal variability of commuting patterns[21] is intertwined with the mode of journey[22], the population density (that is, urbanization level)[23] and the socio-economic status[24–26].

CDR and LBS contribute to the continuous creation of snapshots of citizens' mobility patterns and represent a needed instrument to provide valuable insights on population dynamics in circumstances that urge rapid response[27,28]. They have been used to inform public health policymakers assessing the spread of a disease across the population[29–31]. Recently, during the coronavirus disease 2019 (COVID-19) pandemic, LBS metrics have become a proxy to evaluate the effectiveness and effects of mobility restriction policies enforced by local governments worldwide[32–35]. Using aggregated mobility data, researchers around the world can develop models to study and predict transmission dynamics[36–38], investigate the impact and effectiveness of restriction policies and re-opening strategies[38–47], and analyse the effects of these policies on the local economy, ethnic and socio-economic groups[48–52]. Moreover, coupling the mobility data with the socio-economic and ethnicity groups from the census, it is possible to estimate the socio-economic impact of such restrictions in each different community[53–57].

The majority of the current literature is focusing on the changes in the spatial dimension of mobility during the COVID-19 pandemic, that is, if citizens are changing the patterns of their whereabouts in terms of

[1]Computer Science Department, University of Exeter, Exeter, UK. [2]The Alan Turing Institute, London, UK. [3]Spectus, New York, NY, USA. [4]Federal University of Ceará, Fortaleza, Brazil. [5]Complex Connections Lab, Network Science Institute, Northeastern University London, London, UK. ✉e-mail: riccardo.diclemente@nulondon.ac.uk

magnitude (radius of gyration[58] and/or the location visited[47]). In both cases, using de-identified data from mobile phone users, the authors employ the radius of gyration to assess the spatial differences in mobility. The temporal analyses are restricted to assessing trip duration of changes in spatial mobility over time. These studies do not address synchronized mobility patterns or other temporal aspects of human mobility during the pandemic. In addition, both works were published in the early stages of the pandemic, so mobility changes in the same population during different lockdowns could not be studied. Besides the spatial patterns, human whereabouts follow temporal regularities driven by physiology, natural cycles and social constructs[59]. Few studies have explored these regularities aiming to characterize temporal components and classify people according to weights on these components[59] or to uncover the emergency social phenomena such as the 'familiar strange'[60]. In the context of the pandemic, it was found that morning activity started later, evening activity started earlier and temporal behavioural patterns on weekdays became more similar to weekends[61]. Since urban mobility patterns are built upon the space–time interaction[62], it is vital also to study both dimensions of mobility to shed light on the mechanisms behind the changes in human mobility during the COVID-19 pandemic.

We can assess the space–time interaction[63] of human activities, studying the rhythms of human mobility with the spatial span of the urban whereabouts. The challenge at hand is to disentangle and investigate how each dimension has been reshaped during the pandemic. To assess the changes in the spatial mobility patterns, preserving citizen privacy under the General Data Protection Regulation (more information available at gdpr-info.eu/, accessed on 1 June 2023), we employ the radius of gyration[64,65] as a spatial metric. The radius of gyration was chosen for being a well-known metric applied to measure human mobility[15,19,64–69]. This measure was used during COVID-19 to gauge the general population's compliance with mobility restrictions[38,47,70], inform policymakers on their decisions[38,44,47] and reveal differences in the impact on different socio-economic groups and minorities[71–74]. Besides the regularities in the spatial dimension, human mobility patterns also exhibit a high degree of temporal regularity[64]. These regularities are related to circadian rhythms[59] and commuting for work[75], study[75] or shopping purposes[76], for example. In our work, to gauge the temporal dimension of human mobility, we defined the mobility synchronization metric to quantify the co-temporal occurrence of the daily mobility motifs. It mainly measures the regularities linked to synchronize work schedules, that is, people leaving home around the same time to go to work. Increased synchronized mobility leads to augmented social contact rates, which elevate the risk of transmission of infectious diseases[60]. Hence, mobility synchronization can provide relevant insights to policy markers during the pandemic. Combining these spatio-temporal metrics gives us an idea of how far infectious individuals could potentially travel and how many people they could be in contact with (for example, public transport and office spaces).

In this Article, leveraging LBS data from de-identified mobile phone users who opted-in to anonymous location sharing for research purposes, we study how citizen mobility patterns changed from January 2019 to February 2021 across the United Kingdom. As the pandemic unfolded, we observed changes in the duration and frequency of trips and disentangled how each mobility dimension was affected. At the lifting of each restrictions, the spatial mobility dimension recovered faster than the temporal dimension. The space–time components drift their trends during the second lockdown to finally align back after the third lockdown. Trips are defined in our paper as the event in which a user leaves their geo-fenced home area. For each trip, we registered the total time spent outside before returning home, if the trip included a green area, and the distance travelled.

We coupled the mobility dimensions with the urbanization, unemployment, occupation and income levels from the census at the local authorities level. Rural and urban areas manifest opposite trends.

In rural areas, the lockdowns affect more the spatial dimension, where the locations of human activities are spread apart and consequently the trips are longer (data from National Travel Survey: England 2018, available at gov.uk/government/statistics/national-travel-survey-2018, accessed on 1 June 2023). Meanwhile, in urbanized areas, the synchronicity of the daily activity was dissolved possibly by the rise of asynchronous communing patterns (for example, flexible work hours or rotational/staggered shifts)[77,78].

We observed that, during the pandemic, the unemployment rates affect more the temporal dimension than the spatial one, where high unemployment levels were associated with low-mobility synchronization. Moreover, this effect seems to be tied with the urbanization level of the local authorities. Lastly, we adopt the national statistics socio-economic classification (NS-SEC) as a proxy to gauge the impact of the pandemic on the mobility patterns of income/occupation groups. We noticed that areas with elevated concentration of population on low-income routine occupations had the most substantial reduction in the spatial and temporal dimensions of mobility.

## Results

Throughout this section, we define the spatial dimension of human mobility as the span of the citizen's movement, that is, the length of the trips. This dimension is gauged using the radius of gyration, which quantifies how far from the centre of a user's mobility the visited geographical locations are spread. In the temporal dimension, we are interested in measuring co-temporal events linked to collective, synchronized behaviours. This dimension is estimated with the mobility synchronization metric that represents temporal regularities related to when people tend to leave their residences at regular time period. Our goal is to quantify how containment measures (for example, limited social gatherings, business and schools closures, and home working) affected travel rhythms of the populations. The intervals are identified through the analysis of the strongest frequency components in Fourier spectra of the out-of-home trips time series. For this analysis, we use the trip data aggregated hourly. More information on these metrics is available in Methods.

We analysed LBS data from January 2019 to February 2021 in the United Kingdom. This period includes the three lockdowns announced by the United Kingdom's prime minister (GOV.UK: Coronavirus press conferences, available at gov.uk/government/collections/slides-and-datasets-to-accompany-coronavirus-press-conferences, accessed on 1 June 2023). Given the discrepancies in the implementation of the different lockdown across each country in the United Kingdom, an analysis of the local/regional impact of the pandemic is included in Supplementary Information: Northern Ireland (Supplementary Fig. 7), Scotland (Supplementary Fig. 8), Wales (Supplementary Fig. 9) and England (Supplementary Fig. 6).

Figure 1 depicts the radius of gyration and mobility synchronization trends from the second week of 2020 to the seventh week of 2021. Both metrics have similar trends up to week 18 of 2020 when the radius starts recovering to pre-pandemic levels while the synchronization does not. The recovery in the spatial dimension coupled with the fluctuations in the temporal patterns suggests that, although people gradually started making trips similar to the period before the pandemic, these trips do not present the temporal synchronization observed before. After the third lockdown, we can notice similar trends in the spatial and temporal dimensions as in the period before week 18.

Since mobility synchronization measures the existing time trends, it can estimate the effects of human mobility restrictions policies, such as lockdowns. We can see reductions in the mobility synchronization levels during all three lockdowns. However, the second one produced the most considerable decrease in the synchronicity levels in most local authorities compared with the other two; 78.34% of the local authorities experienced a reduction in the synchronization level compared with the baseline. In contrast, during the first and the third lockdown,

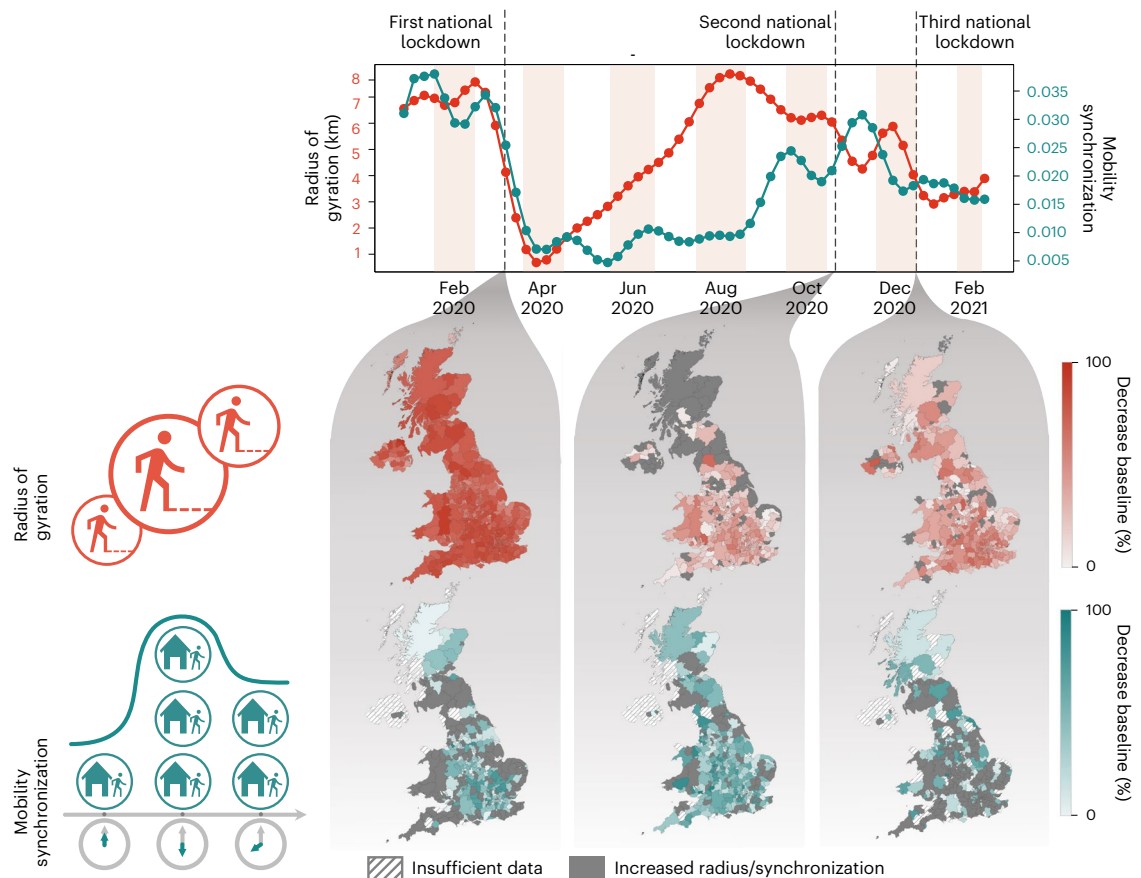

**Fig. 1 | Evolution of the radius of gyration and mobility synchronization in the United Kingdom's local authorities from the second week of 2020 to week 7 of 2021.** The maps depict the effects of the three English national lockdowns regarding the spatio-temporal metrics adopted. Note that the first lockdown resulted in the most notable reduction in the radius of gyration in all four nations. However, it is worth mentioning that the first lockdown was the only one with the same restrictions for all of the United Kingdom's countries. Concerning mobility synchronization, the most notable reduction occurred during the second lockdown. Additionally, it is challenging to disentangle the effects of the third lockdown from the changes in the population mobility patterns caused by end-of-year holidays.

the mobility synchronization decreased in 56.67% and 34.02% of the local authorities, respectively. The substantial drop in the mobility synchronization during the second lockdown might be tied to the changes in the stay-at-home policy[79–81]. This policy mainly only allowed essential workers to leave home to work, while in the second and third lockdowns this rule became more flexible and people who could not work from home were allowed to go to work[79,82–85].

### Changes in mobility according to urbanization level and economic stratification

Human mobility patterns are also affected by urbanization level[86]. For example, rural areas are characterized by limited accessibility to goods, services and activities[87]. In the context of a pandemic, the urbanization level partly explains differences in mobility patterns[88], and the diffusion of infectious diseases[89–91].

To assess how the spatio-temporal mobility patterns of areas with different levels of urbanization were affected during the COVID-19 pandemic, we analyse the radius of gyration and mobility synchronization by urbanization level. We divided the local authorities into three classes accordingly to the urban–rural classification adopted for England[92] illustrated in Fig. 2a.

We employed the concept of residual activity[64] to visualize the deviations of the mobility patterns during the lockdowns when compared with their expected behaviour (for example, we used the same period of 2019 as the baseline for comparisons). Figure 2b depicts the different responses of the urban–rural group to the three national lockdowns. High residual values indicate an increased number of trips compared with the expected behaviour (baseline period).

During the first lockdown, urban areas presented an increase in expected mobility. In contrast, rural areas have a negative trend. However, during the second and third lockdown, an opposite scenario emerges. Rural local authorities increased the expected residual activity, and urban areas decreased it. These differences can be driven by the change in the mobility restriction policies (for example, more flexible stay-at-home rules[79]) and the characteristics and pre-existing social vulnerabilities of urban and rural areas, as found in previous works[93].

Although there is a debate as to whether a high population density accelerates or not the spread of the virus[94], other urban and rural characteristics can be risk factors for COVID-19. For example, transportation systems and increased inter-/intra-urban connectivity are regarded as key factors contributing to the spread of contagious diseases[95].

The results of the radius of gyration (Fig. 2c) and the mobility synchronization (Fig. 2d) also indicate differences in the response of urban and rural areas to the lockdowns. Due to the characteristics of the geographic distribution of local amenities in rural areas, people tend to have a greater radius of gyration compared with urban areas (data from National Travel Survey: England 2018, available at gov.uk/government/statistics/national-travel-survey-2018, accessed on 1 June 2023).

Analysing the trends of the radius of gyration and mobility synchronization compared with the baseline of 2019, we can notice some

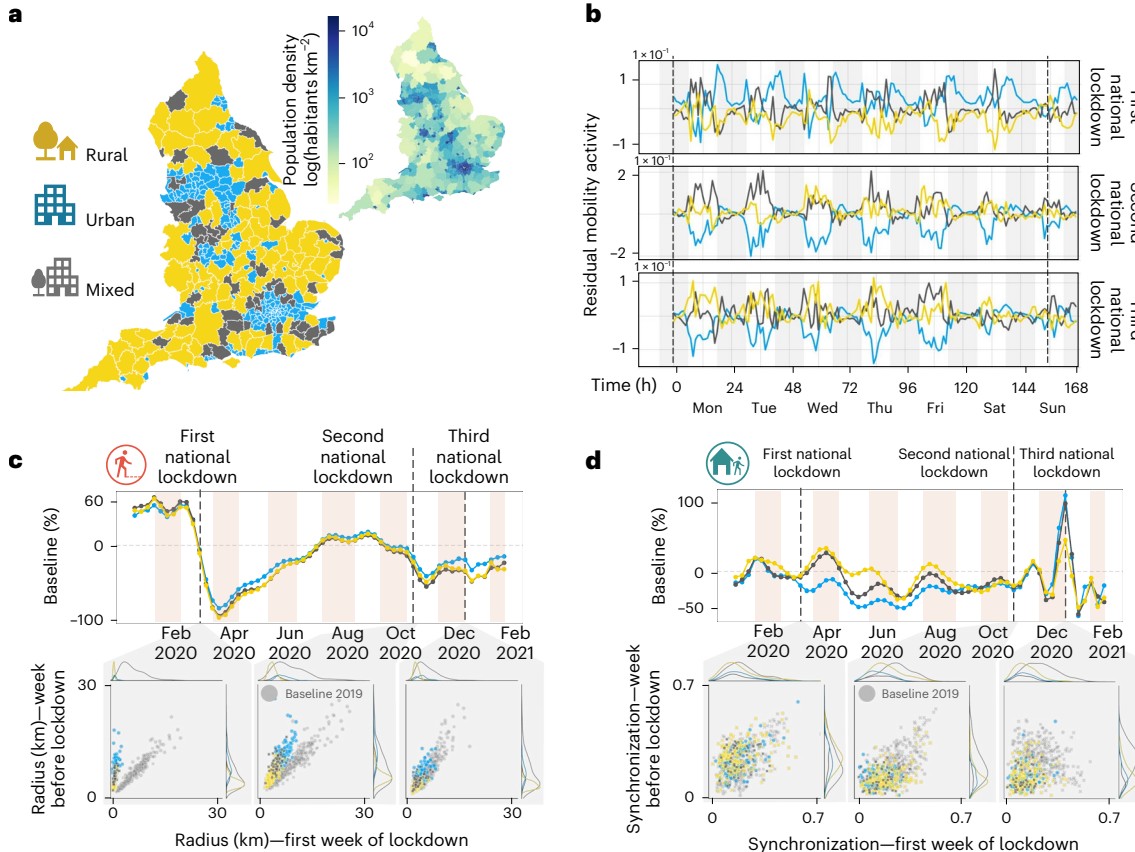

**Fig. 2 | Radius of gyration and mobility synchronization of English local authorities grouped according to the urban–rural classification. a**, The urban–rural classification of the English local authorities. **b**, The differences in the number of out-of-home trips for the different urban–rural groups.

Notice that, after the first lockdown, urban local authorities started to present negative values in their curve. **c**,**d**, Time series with variations on the radius of gyration (**c**) and mobility synchronization (**d**) compared with the baseline year (2019).

differences in the urban–rural and spatio-temporal response of the local authorities (scatter plots in Fig. 2c,d). In the spatial dimension, we can see the same behaviour for urban and rural areas, characterized by a reduction in the mobility levels in the week before and the first week of lockdown. The shift between the baseline and the first/second lockdown indicates that the radius during the first week was substantially smaller than the week before these lockdowns.

In the temporal dimension, however, the differences between the mobility levels in the week before the lockdown and the first week of lockdown (first scatter plot in Fig. 2d) are less dramatic than observed in the spatial dimension. Nonetheless, compared with the baseline, we can still see a reduction in the synchronization values for urban and rural areas, especially in the second and third lockdowns (baseline plot in Fig. 2d).

As discussed, the number of trips has decreased across all the urban–rural groups during the pandemic. Since work-related activities often create the necessity to leave home, the rise in home working and unemployment rate contributes to the reduction in the mobility levels[96]. Next, we study the relationship between the spatio-temporal mobility metrics and the unemployment rate for areas with different levels of urbanization, both before and during the pandemic. We estimate the size of the unemployed population based on the unemployment claimant count (data from the Office for National Statistics (ONS), available at ons.gov.uk/employmentandlabourmarket, accessed on 1 June 2023).

We divided the time series of the radius of gyration and mobility synchronization into pre-pandemic (from April 2019 to February 2020)

and pandemic (from April 2020 to February 2021) periods, and we analysed the Kendall Tau correlation between them and unemployment claimant count.

At first glance, the positive correlation between the radius of gyration and the unemployment rate (Fig. 3a,b) seems to be dissonant from previous works[25,97]. However, this result can be due to a rise in the unemployment rate and the spatial mobility levels before the pandemic. For the pandemic period, although the lockdowns have reduced the radius during specific periods, we see in Fig. 1 a period between April and August 2020 when the radius increased, making the correlation positive.

While the correlation between the spatial dimension of mobility and unemployment drops only marginally, preserving the positive sign and affecting more urban than rural areas, the correlation between the time dimension of mobility and unemployment drops considerably, becoming negative and impacting more rural districts than the urban (Fig. 3c,d).

The same explanation applies to mobility synchronization in the pre-pandemic period. During the pandemic, we can see that the spatio-temporal dimensions display different patterns between May and November, and the temporal one does not present the same steep recovery as the spatial between April and August 2020. The oscillations in the patterns of the temporal dimension impacted the correlation with unemployment, making it negative.

We argued that work trips contribute to the creation of our mobility patterns. Moreover, the type of occupation, among other socio-demographic characteristics, also influences those patterns[98].

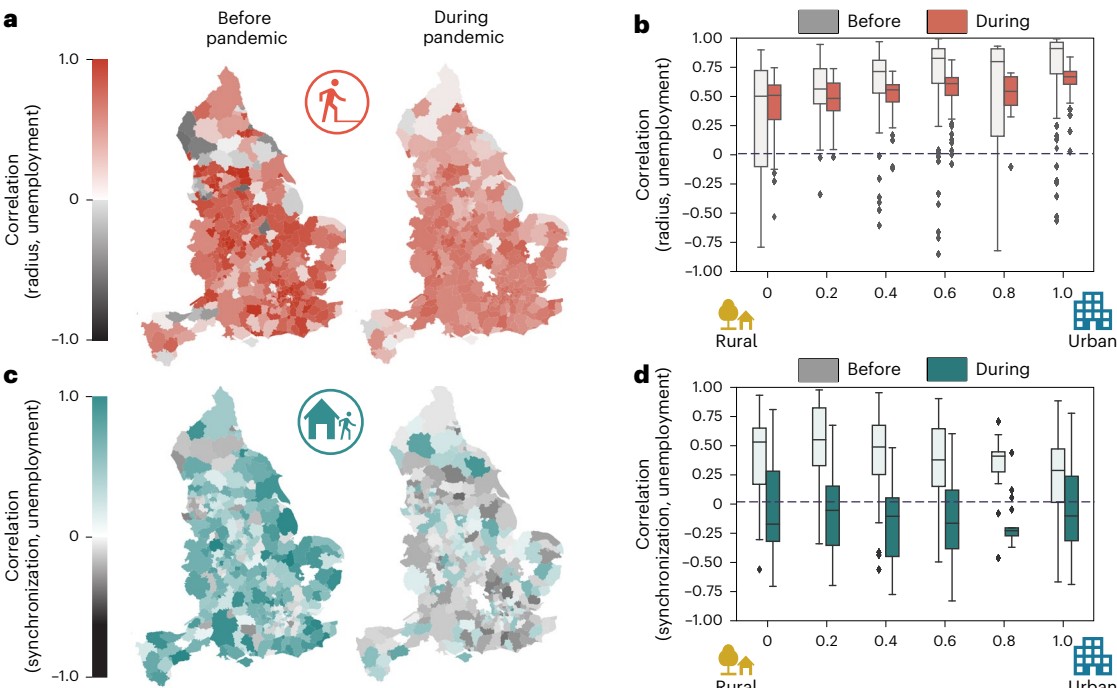

**Fig. 3 | Correlation between the spatio-temporal metrics of human mobility and the unemployment claimant count. a**, The correlation between the radius of gyration of the English local authorities ($N$ = 316 English local authorities) is depicted in before and during the pandemic. **b**, The correlation when the local authorities are grouped by level of urbanization also before and during the pandemic. **c**, The correlation between mobility synchronization with the unemployment claimant count of the English local authorities ($N$ = 316 English local authorities) for the periods before and during the pandemic ($N$ = 316 English local authorities). **d**. The results grouped by urbanization level. We define the period before the pandemic from April 2019 to February 2020. The pandemic period considered was from April 2020 to February 2021. In **b** and **d**, boxes that span from the 25th percentile (Q1) to the 75th percentile (Q3), with median values (50th percentile) represented by a central line. The minimum and maximum values are determined by subtracting 1.5 times the interquartile range (IQR) from Q1 and adding 1.5 times the IQR to Q3, respectively. Notice that, before the pandemic, both the radius and the synchronization positively correlated with the unemployment claimant count. However, during the pandemic, the correlation with the radius became less strong, and that with synchronization became negative. It is also worth mentioning that **b** and **d** reveal a possible association between correlation changes and areas' urbanization level.

In this sense, we use the English NS-SEC (available at ons.gov.uk, accessed on 1 June 2023) to gauge the response of employment relations and conditions of occupations to the pandemic. Using this classification, we also have insights into the connection between the spatio-temporal mobility and wealth level. This analysis is important since wealth and economic segregation are linked to differences in mobility patterns and response to human mobility restrictions under the COVID-19 pandemic[33].

In the NS-SEC classification, classes related to managerial occupations exhibit strong positive correlation with income as depicted in Fig. 4a. In contrast, lower supervisory, technical, semi-routine or routine occupations negatively correlate with income. The remainder of NS-SEC classes present a strong correlation with income.

Moreover, Fig. 4b shows that, before and during the COVID-19 pandemic, people's radius of gyration and the mobility synchronization within an area tend to grow as the concentration of population in managerial occupations increases. The opposite result is observed when analysing the percentage of the population (quartiles Q1 to Q4) in lower supervisory, semi-routine or routine occupations.

In all scenarios depicted in Fig. 4, we can see a reduction in the temporal and spatial dimensions of mobility during the pandemic. However, each group contributed differently to the overall change in the mobility observed during the pandemic. As mentioned before, the first national lockdown produced the most substantial impact on the spatial dimension of mobility. In contrast, the second greatly impacted the temporal dimension.

Besides the difference in the time–space facets of human mobility analysed, changes in the duration[99] and the purpose[100] of the trips were also observed during the pandemic. Researchers found that these changes vary accordingly to income levels[100] and could be related to the emergence of new habits[101].

To assess the changes in the duration of the trips, we measured the time elapsed when the user left their home geo-fencing area and entered it again. We can further disaggregate the trips by classifying them as work-related and other types based on their starting time. Comparing the duration of the trips in a week with no mobility restriction Fig. 5a with a week with lockdown Fig. 5b, we can see that trips classified as work-related display a reduction in their length. The analysis based on the income/socio-economic groups shows that high-income groups presented the most notable reduction compared with the baseline year of 2019, depicted in Fig. 5b. This result is in line with a similar paper in the literature, which also reports differences related to the income groups[61,99].

Besides the trips' duration, another relevant aspect being analysed is the impact on the type of trips. We study the differences in leisure-related trips before and during the pandemic to assess this impact. Here these trips are estimated on the basis of examining trips that include green areas such as parks, sports facilities and play areas. In a period without any mobility restriction measures, we expect to see an increase in these trips from the end of winter until the end of summer. This number should start decreasing as when the winter approaches. As shown in Fig. 5d, using the baseline year of 2019, before the first national lockdown, there was no substantial difference between the number of trips by rural and urban local authorities. This pattern stays consistent until the start of the summer when the restrictions are stated to be lifted[79]. After that, we can see a substantial difference in the amount of

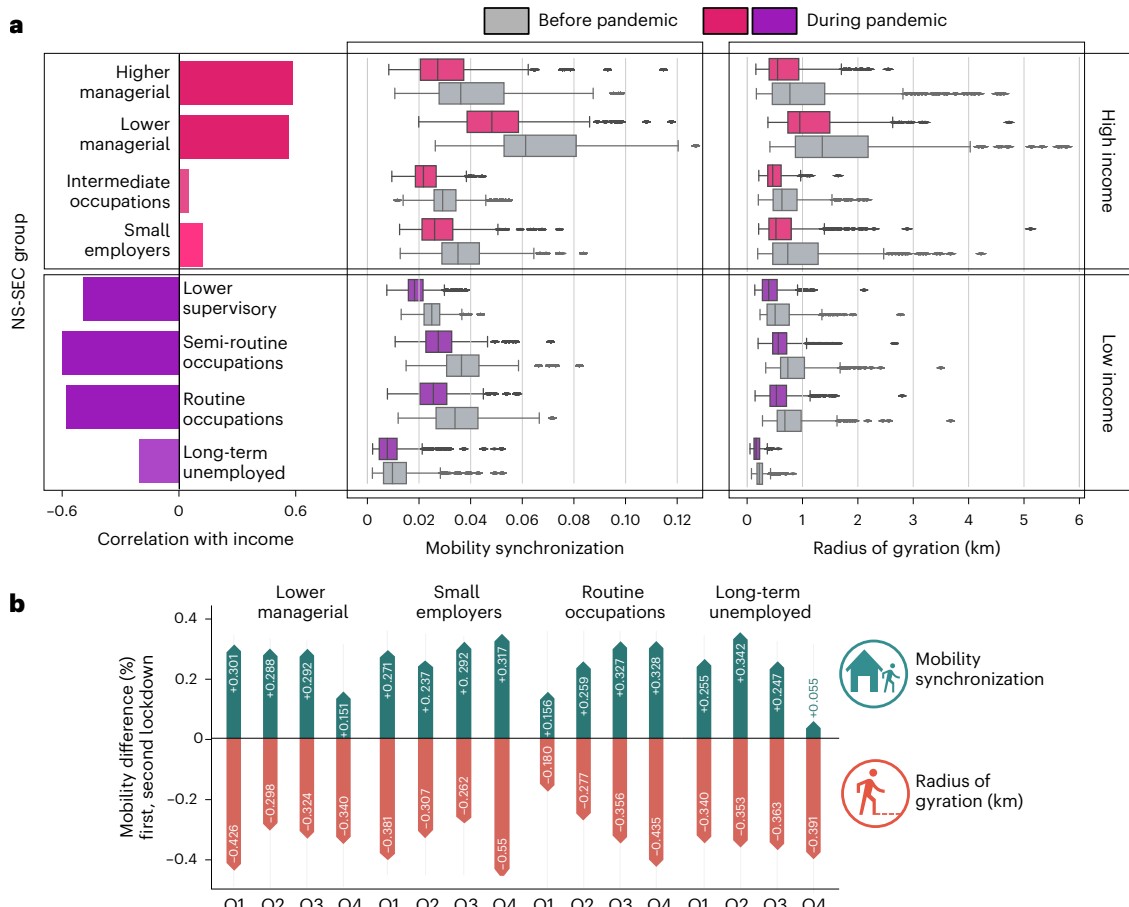

**Fig. 4 | Relation between the NS-SEC classification, income and their impact on the spatio-temporal mobility patterns before and during the pandemic.** The 'Before pandemic' label corresponds to the period from April 2019 to February 2020, while the 'During pandemic' corresponds to the period between April 2020 and February 2021. **a**, The correlation between the NS-SEC classes and the population income ($N = 316$ English local authorities). The classes coloured in pink and purple have positive and negative correlations with the population income. The box plot consists of boxes that span from the 25th percentile (Q1) to the 75th percentile (Q3), with median values (50th percentile) represented by a central line. The minimum and maximum values are determined by subtracting 1.5 times the interquartile range (IQR) from Q1 and adding 1.5 times the IQR to Q3, respectively. Also shown is the radius of gyration/mobility synchronization of each NS-SEC class before and during the pandemic. **b**, The differences between the first and second lockdowns in the spatio-temporal metrics.

## Discussion

The accurate estimation of spatio-temporal facets of human mobility and gauging its changes as a reflection of the pandemic and mobility restriction policies is critical for assessing the effectiveness of these policies and mitigating the spread of COVID-19 (refs. 79,82–84,102). A vital contribution of our work lies in applying two metrics to disentangle the changes in mobility's temporal and spatial dimensions. Using the radius of gyration, we could identify that the effect of the first lockdown was more substantial than the other ones in changing the spatial characteristics of citizens' movement. Among the reasons that could lead to this result, we can mention more strict policies adopted in the first lockdown and the lockdown duration[79,85]. However, further investigation is needed to obtain more pieces of evidence to support these hypotheses.

In contrast to the spatial dimension of mobility, the results indicate that the temporal one was more impacted during the second lockdown when more flexible mobility restriction policies were enforced. After the first lockdown, we argue that people who could not work from home were allowed to leave home and work in the office as long as they

respected social distancing rules[81,85]. Different work shifts were created to comply with these rules and limit the number of people inside indoor spaces. As a result, instead of having the same, or similar, work schedule for all employees, they started to be divided into groups that should go on different days of the week and at different times, impacting the temporal synchronization of their movement.

Besides the changes during different periods and under different mobility restriction policies, we also analysed the interplay between the characteristics of the area (for example, level of urbanization, income and unemployment rates) and the spatio-temporal human mobility metrics adopted. We noticed that rural areas presented a more considerable reduction in spatial patterns compared with urban ones. At the same time, urban areas were more impacted in their temporal dimension than rural ones. These differences observed in the urban–rural classification are correlated with the population density of the regions and can affect the impact of the mobility restriction measures.

We also observed changes in the response of regions regarding their unemployment rates and populations income. For the unemployment rates, we observed that before the COVID-19 pandemic, the unemployment claimant count was positively correlated with the radius of gyration and mobility synchronization in the majority of the areas. However, during the pandemic, this correlation became weaker for the radius of gyration and negative for mobility synchronization.

green areas trips in rural and urban areas. Although one could argue that this could indicate a preference towards more natural leisure trips in rural areas, further investigation is required to identify the reasons behind these differences.

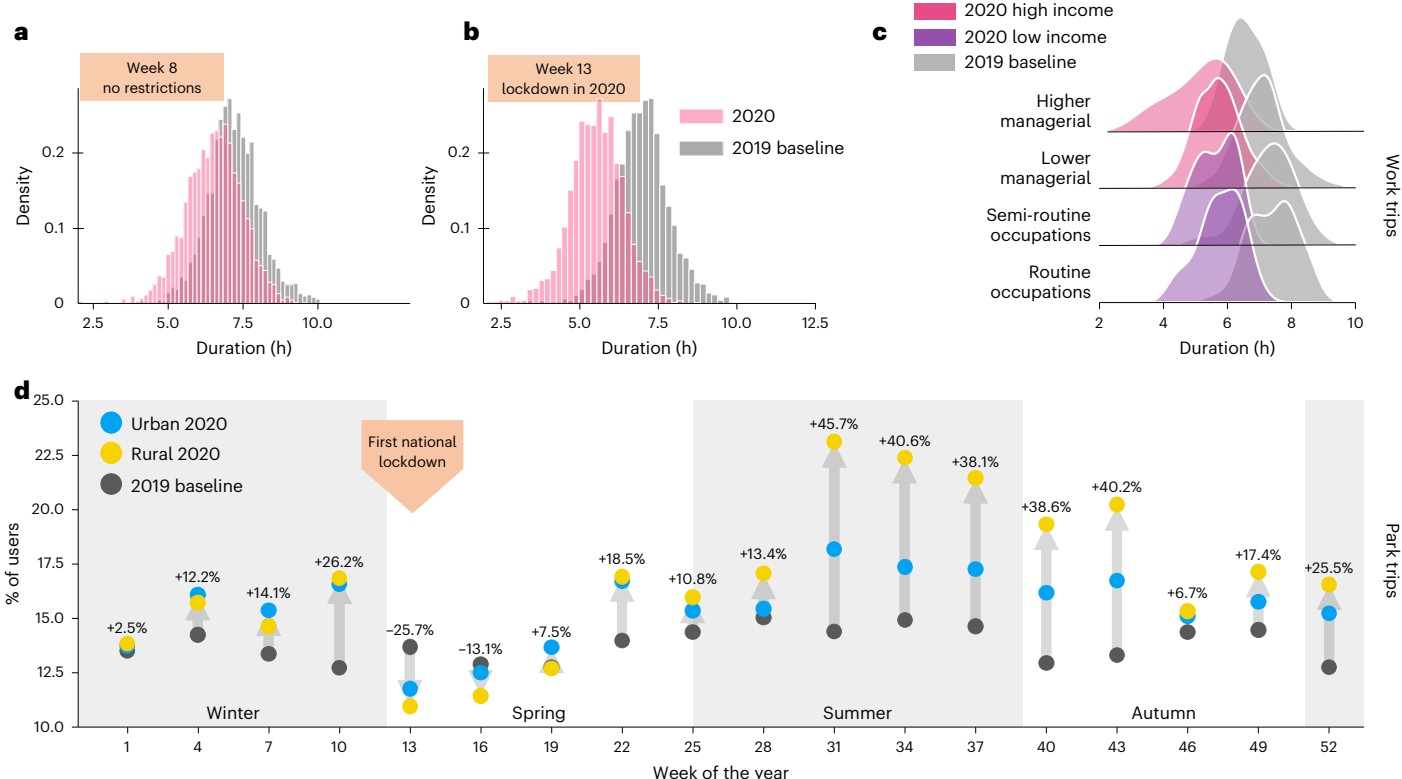

**Fig. 5 | Changes in the number and duration of trips. a,b,** The differences between the duration of work-related trips for 2 weeks in 2019 and 2020. During week 8 (**a**), there were no mobility restrictions in 2020, and we can see that the distributions are similar. In contrast, week 13 (**b**) was the first week of the first lockdown, and we can see more differences in the distributions of 2019 and 2020. **c,** The differences across different socio-economic groups during week 13. **d,** The differences in the number of trips to green spaces such as parks in 2020 compared with the baseline year of 2019. Note that, in this case, the local authorities are grouped in urban and rural groups.

Less urbanized areas tend to have a lower spatial correlation and a higher temporal correlation with unemployment than more urbanized areas. Using the NS-SEC classification as a proxy to assess the response of different income and work groups, we observed that low-income routine/semi-routine occupations were the groups that presented the greatest reduction in their radius and synchronization. Moreover, the changes in the mobility restriction policies after the first lockdown had an impact on these groups, which can be the reason behind the greatest impact on the temporal dimension of mobility[79,85]. Similarly, changes were also observed concerning the duration of work-related trips and the number of trips to green areas. These differences were also observed at urbanization and socio-economic levels.

Regarding our work's contribution to policymakers, we argue that the spatio-temporal metrics employed in this study help to assess mobility changes before and after the implementation of policies, as seen in the first and second lockdowns with flexible stay-at-home policies[79–81]. Moreover, our results indicate that different groups (socio-economic and urban–rural) experience and respond to these policies differently. These results were also found by similar works[74,93,96,99] and provide insight to policymakers to design strategies that consider each group's particularities.

In summary, the analysis of the spatial dimension of human mobility coupled with the insights from the study of the temporal dimension allows us to characterize the impact of policies such as stay-at-home and school closures on the population of different areas/socio-economics. These differences suggest that each group experiences, in a particular way, the emergence of asynchronous mobility patterns primarily due to the enforcement of mobility restriction policies and new habits (for example, home office and home education).

## Methods

### Data sources

**Human mobility data.** Spectus provided the human mobility data used in this research for research purposes. These data were collected from anonymous mobile phone users who have opted-in to give access to their location data anonymously, through a General Data Protection Regulation-compliant framework. Researchers queried the mobility data through an auditable, cloud-hosted sandbox environment, receiving aggregate outputs in return. The datasets contain records of UK users from January 2019 to early March 2021. In total, we have over 17.8 billion out-of-home trips and about 1 billion users' radius of gyration records. Note that the radius logs are measured on a weekly basis, while the trips are recorded on a daily basis. More information on the datasets is available in Supplementary Information. We assessed the representativeness of the data by analysing the correlation between the number of users and the population of the local authorities. A strong positive correlation between the populations compared with $r^2$ value equal to 0.775 was obtained (Supplementary Fig. 1a). The data are composed of records of when users leave the area (as a square of 500 m) that is related to their homes and the length of the trips. The home area is estimated on the basis of the history of places visited, the amount of time spent at the area and the period when the user stayed at the location following[21,103]. It is worth mentioning that we only used mobility-related data in this work due to privacy concerns. No personal information or another type of information that would allow the identification of uses was used. Also, the data are aggregated at the local authority level as the aim is to study trends at the population level rather than the individuals. The dataset on the trips to the green areas is composed of records of the number of trips that include green spaces

across Great Britain. These records were collected daily grouped by the hour the trip starts, and are aggregated at the local authority level.

**Other data sources.** The Income study employed data from the report on income estimates for small areas in England in 2018 provided by the ONS[104]. It is available for download and distribution under the terms of the Open Government Licence (available at www.nationalarchives. gov.uk/, accessed on 1 June 2023). Similarly, the analysis of the unemployment claimant rate, the urban–rural classification of English local authorities[92] and the NS-SEC[105] used the data published by the ONS publicly available under the terms of the Open Government Licence. The remaining socio-economic data utilized aggregated data from the UK Census of 2011 (ref. 106), which is available for download at the InFuse platform also under the terms of the Open Government Licence. For the green area study, we used the Ordnance Survey Open Greenspace dataset to obtain information on the locations of public parks, playing fields, sports facilities and play areas (OS Open Greenspace, available at ordnancesurvey.co.uk/os-open-greenspace, accessed on 1 June 2023). Our analysis did not include categories related to religious grounds, such as burial grounds or churchyards.

## Metrics and other methods

**Radius of gyration.** We conducted the study of the radius of gyration (RG) following the definition of Gonzalez et al.[64]. It can be described as the characteristic distance travelled by a user $u$ during a period and is calculated as follows

$$RG_u = \sqrt{\frac{1}{N_u} \sum_{i=1}^{N_u} (\vec{r}_u^{\,i} - \vec{r}_u^{\,cm})^2}$$ (1)

where, $N_u$ represents the unique locations visited by the user, $\vec{r}_u^{\,i}$ is the geographic coordinate of location $i$ and $\vec{r}_u^{\,cm}$ indicates the centre of mass of the trajectory calculated by

$$\vec{r}_u^{\,cm} = \frac{1}{N_u} \sum_{i=1}^{N_u} n_u^i \vec{r}_u^{\,i}$$ (2)

where $n_u^i$ is the visit frequency or the waiting time in location $i$. The mobility value of each region is the median value of the radius of gyration of the users within a temporal window of 8 days centred around a given day.

**Residual mobility activity.** The concept of residual mobility activity displayed in Fig. 2b was adapted from ref. 107, and it is used to highlight differences between the measured behaviour of the local authorities compared with their expected behaviour. For a given local authority $i$ is calculated as follows

$$a_i^{res}(t) = a_i^{norm}(t) - a^{-norm}(t)$$ (3)

where $a^{-norm}(t)$ is the normalized activity averaged over all local authorities under at each particular time, and $a_i^{norm}(t)$ is computed similarly to the $Z$-score metric

$$a_i^{norm}(t) = \frac{a_i^{abs}(t) - \mu^{abs}}{\sigma^{abs}}$$ (4)

where $a_i^{abs}(t)$ is the activity in a local authority at a specific time $t$, $\mu^{abs}$ is the mean activity of all local authorities under the same urban–rural classification of $i$ at a specific time, and $\sigma^{abs}$ represents the standard deviation of all local authorities under the same urban–rural classification of $a_{i,j}$ at a particular time.

**Mobility synchronization.** The mobility synchronization is not limited to conventional commuting routines. It can happen at any time of the

day in which people tend to perform certain activities. For example, school teachers, healthcare professionals and other routine or semi-routine occupations tend to have defined times reserved for specific activities (for example, eating, exercising and socializing). To have a more accurate portrait of the mobility synchronization patterns, instead of analysing it as a concentration of trips around certain hours, we define the mobility synchronization as the total magnitude in the periodicity in the out-of-home trips.

First, using 2019 as a baseline, we analyse the wavelet and Fourier spectra to determine the expected strongest frequency components in the mobility regularity. For a given mother wavelet $\psi(t)$, the discrete wavelet transform can be described as

$$\psi_{j,k}(t) = \frac{1}{\sqrt{2^j}} \psi\left(\frac{t - k2^j}{2^j}\right)$$ (5)

where $j$ and $k$ are integers that represent, respectively, the scale and the shift parameters. For the Fourier transform, a discrete transform of the signal $x_n$, for $n = 0...N - 1$ is:

$$X_k = \sum_{n=0}^{N-1} x_n e^{-i2\pi kn/N}$$ (6)

where $K = 0...N - 1$ and $e^{-i2\pi kn/N}$ represents the $N$th roots of unity.

Employing these two transforms, we found that the mobility patterns are characterized by five main periods, namely 24 h, 12 h, 8 h and 6 h (Supplementary Fig. 3). However, because the 24 h component overshadows the other three components (Supplementary Fig. 5), we focus the analysis on the 12 h, 8 h and 6 h periods. Moreover, during the pandemic, these periods were more affected than the 24 h component (Supplementary Fig. 4). Next, the mobility synchronization metric is defined as the sum of the powers from the Lomb–Scargle periodograms[108] corresponding to the 12 h, 8 h and 6 h. The generalized Lomb–Scargle periodograms is calculated as

$$P_N(f) = \frac{1}{\sum_i y_i^2} \left\{ \frac{\left[\sum_i y_i \cos\omega(t_i - \tau)\right]^2}{\sum_i \cos^2\omega(t_i - \tau)} + \frac{\left[\sum_i y_i \sin\omega(t_i - \tau)\right]^2}{\sum_i \sin^2\omega(t_i - \tau)} \right\}$$ (7)

where $y_i$ represents the $N$ measurements of a time series at time $t_i$, $\omega$ is frequency and $\tau$ can be obtained from

$$\tan 2\omega\tau = \frac{\sum_i \sin 2\omega t_i}{\sum_i \cos 2\omega t_i}$$ (8)

The mobility synchronization is a value between 0 and 1, where higher values for a given period indicate that more people left their homes simultaneously. In the context of the pandemic, high mobility synchronization can be translated into a potential increase in the likelihood of being exposed or exposing more people to the virus due to the large number of people moving simultaneously.

## Reporting summary

Further information on research design is available in the Nature Portfolio Reporting Summary linked to this article.

## Data availability

The paper contains all the necessary information to assess its conclusions, including details found in both the paper and Supplementary Information. Due to contractual and privacy obligations, we are unable to share the raw mobile phone data. However, access can be provided by Spectus upon agreement and signature of the non-disclosure agreement. More information on data access for research can be found at Spectus -"Data for Good" movement.

## Code availability

Scripts and Notebooks in Python with our analyses and to reproduce the results in this paper were archived with Zenodo (https://doi.org/10.5281/zenodo.8014785).

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

## Acknowledgements

This work is a collaboration between the Department of Computer Science of the University of Exeter and Spectus in response to the COVID-19 crisis. Spectus is providing insights to academic and humanitarian groups through a multi-stakeholder data collaborative for timely and ethical analysis of aggregate human mobility patterns. This manuscript is the outcome of this collaboration with Spectus and summarizes the main findings. Besides this paper, two additional, non-peer-reviewed, reports were produced during the first national lockdown with an initial analysis of the pandemic in the United Kingdom and a second one analysing changes in socio-economic aspects related to mobility patterns in the United Kingdom during the first lockdown. Both reports are available at the project web page covid19-uk-mobility.github.io. R.D.C. acknowledges Sony CSL Laboratories in Paris for hosting him during part of the research. F.B. was funded by the Economic and Social Research Council (ESRC) & ADR UK as part of the ESRC-ADR UK No.10 Data Science (10DS) fellowship (grant number ES/W003937/1). The funders had no role in study design, data collection and analysis, decision to publish or preparation of the manuscript.

## Author contributions

C.S. performed the analysis. R.D.C. and F.P. gathered the data. R.D.C., F.B., H.B. and R.M. designed the analysis. C.S. and R.D.C. wrote the paper. R.M. and R.D.C. supervised the project. All authors discussed the results and contributed to the final manuscript.

## Competing interests

The authors declare no competing interests.

## Additional information

**Correspondence and requests for materials** should be addressed to Riccardo Di Clemente.

| | |
|---|---|

# Reporting Summary

## Statistics

For all statistical analyses, confirm that the following items are present in the figure legend, table legend, main text, or Methods section.

| n/a | Confirmed | |
|---|---|---|
| ☐ | ☒ | The exact sample size ($n$) for each experimental group/condition, given as a discrete number and unit of measurement |
| ☐ | ☒ | A statement on whether measurements were taken from distinct samples or whether the same sample was measured repeatedly |
| ☒ | ☐ | The statistical test(s) used AND whether they are one- or two-sided<br>*Only common tests should be described solely by name; describe more complex techniques in the Methods section.* |
| ☐ | ☒ | A description of all covariates tested |
| ☐ | ☒ | A description of any assumptions or corrections, such as tests of normality and adjustment for multiple comparisons |
| ☐ | ☒ | A full description of the statistical parameters including central tendency (e.g. means) or other basic estimates (e.g. regression coefficient) AND variation (e.g. standard deviation) or associated estimates of uncertainty (e.g. confidence intervals) |
| ☒ | ☐ | For null hypothesis testing, the test statistic (e.g. $F$, $t$, $r$) with confidence intervals, effect sizes, degrees of freedom and $P$ value noted<br>*Give P values as exact values whenever suitable.* |
| ☒ | ☐ | For Bayesian analysis, information on the choice of priors and Markov chain Monte Carlo settings |
| ☒ | ☐ | For hierarchical and complex designs, identification of the appropriate level for tests and full reporting of outcomes |
| ☒ | ☐ | Estimates of effect sizes (e.g. Cohen's $d$, Pearson's $r$), indicating how they were calculated |

*Our web collection on statistics for biologists contains articles on many of the points above.*

## Software and code

Policy information about availability of computer code

| | |
|---|---|
| Data collection | Spectus Inc provided the data used in this research. No additional softer fer data collection was employed. |
| Data analysis | All software used and implemented for the analysis are free and open source. They will be available in the project GitHub page after the publication (covid-uk-mobility.github.io). This information is described in the code availability section of the main manuscript. |

For manuscripts utilizing custom algorithms or software that are central to the research but not yet described in published literature, software must be made available to editors and reviewers. We strongly encourage code deposition in a community repository (e.g. GitHub). See the Nature Portfolio guidelines for submitting code & software for further information.

## Data

Policy information about availability of data

All manuscripts must include a data availability statement. This statement should provide the following information, where applicable:
- Accession codes, unique identifiers, or web links for publicly available datasets
- A description of any restrictions on data availability
- For clinical datasets or third party data, please ensure that the statement adheres to our policy

The dataset with human mobility information was collected and provided by Spectus Ins, and for contractual reasons, it can not be published. The income study employed data from the report on income estimates for small areas in England in 2018 provided by the Office for National Statistics (ONS). Similarly, the unemployment claimant rate analysis used the data published by the ONS, publicly available under the terms of the Open Government Licence. For the green area study, we used the Ordnance Survey Open Greenspace dataset to obtain information on the locations of public parks, playing fields, sports facilities and play areas (OS Open Greenspace). Lastly, the local authorities' income and other socioeconomic indicators were based on data from the UK Census of 2011, which is available for download at the InFuse platform under the terms of the Open Government Licence.

# Field-specific reporting

Please select the one below that is the best fit for your research. If you are not sure, read the appropriate sections before making your selection.

☐ Life sciences ☒ Behavioural & social sciences ☐ Ecological, evolutionary & environmental sciences

For a reference copy of the document with all sections, see nature.com/documents/nr-reporting-summary-flat.pdf

# Behavioural & social sciences study design

All studies must disclose on these points even when the disclosure is negative.

| | |
|---|---|
| Study description | This is a quantitative study based on quantitative data sources. Both the dataset with the human mobility information and the other data sources (e.g. census, ONS and Ordnance Survey data) are composed of quantitative aspects of the population studied. We used location-based data from de-identified mobile phone users (17.8 billion out-of-home trips and about 1 billion radius of gyration records) from January 2019 to early March 2021 to capture the differences in space-time mobility patterns. We observed that a decrease in spatial mobility (gauged with the radius of gyration) is interwoven with the emergence of asynchronous mobility dynamics (computed using the mobility synchronisation metrics) during lockdown periods. Coupling these results with quantitative census data, we noticed that in less urbanised and low-income areas, the spatial mobility dimension had a more significant disruption compared to the urbanised and high-income areas. In contrast, the opposite situation was observed in the temporal dimension. |
| Research sample | The sample studied is composed of mobile phone users in the UK who opted-in to share their information. The UK sample was chosen due to the need for studies concerning human mobility patterns in its population during the COVID-19 pandemic, the availability of the dataset with records of mobility patterns and additional socioeconomics/demographic indications. Our sample comprised about 1 billion users' radius of gyration records and more than 17.8 billion out-of-home trips. The human mobility data was provided by Spectus Inc and was aggregated by the local authority level due to privacy concerns. The radius of gyration data was measured weekly, while the trips were recorded daily. The Census, ONS and Ordnance survey datasets are publically available online and were also aggregated at the local authority level. we conducted experiments to ensure that our sample of the population was representative of the local authorities' total population, as can be seen in the SI information of the paper. |
| Sampling strategy | For all the experiments, we used all UK users in the dataset provided by Spectus Inc. Since these users are associated with a geographic area, they could be mapped to a local authority based on their estimated home location (this procedure is explained in trials in our paper). For the green areas, urbanisation and socioeconomics analyses, we limited our study to the local authorities in England to guarantee the consistency of the data (the same type of data was not available in all four UK countries or was collected using different methodologies.). We performed verifications to ensure that our user base was representative of the population of the local authorities, and the results are illustrated in the supplementary information. |
| Data collection | The human mobility data were collected from anonymous mobile phone users who have opted-in to give access to their location data through a GDPR-compliant framework. This collection was made automatically by an application which sends the data to the company's servers, from where researchers could query the data through an auditable, cloud-hosted sandbox environment. The platform outputs only aggerated information. The remainder dataset was collected from the Office for National Statistics website (income and urbanisation data), Ordnance Survey (green areas data), and InFuse platform (census 2011 data). |
| Timing | From January 2019 to March 2021 |
| Data exclusions | No data was excluded from the analyses. |
| Non-participation | The human mobility data users that removed their location data permission were automatically removed from the sandbox environment. Due to contractual reasons, we can not disclose that exact number. However, we conducted experiments to ensure that our sample of the population was representative of the local authorities' total population, as can be seen in the SI information of the paper. For the census data, there were 31 million men and 32.2 million women in the UK. The estimated populations of the four constituent countries of the UK are 53 million people in England, 5.3 million in Scotland, 3.1 million in Wales and 1.8 million in Northern Ireland. |
| Randomization | Due to the nature of the studies conducted, randomisation was optional. The uses were assigned to a local authority based on their home location. The classification of the local authorities according to their urbanisation level and income/unemployment was based on data published by the Office for National Statistics. The visits to green spaces were gauged based on the location records of the user's device. Lastly, the analyses of the spatial-temporal patterns of human mobility were based on the location records and did not require user randomisation. |

# Reporting for specific materials, systems and methods

We require information from authors about some types of materials, experimental systems and methods used in many studies. Here, indicate whether each material, system or method listed is relevant to your study. If you are not sure if a list item applies to your research, read the appropriate section before selecting a response.

## Materials & experimental systems

| n/a | Involved in the study |
|-----|----------------------|
| ☒ | Antibodies |
| ☒ | Eukaryotic cell lines |
| ☒ | Palaeontology and archaeology |
| ☒ | Animals and other organisms |
| ☒ | Human research participants |
| ☒ | Clinical data |
| ☒ | Dual use research of concern |

## Methods

| n/a | Involved in the study |
|-----|----------------------|
| ☒ | ChIP-seq |
| ☒ | Flow cytometry |
| ☒ | MRI-based neuroimaging |

