## [Peer Review File · Nature Human Behaviour]

Peer Review Information

Journal: Nature Human Behaviour

Manuscript Title: COVID-19 is linked to changes in the time-space dimension of human mobility

Corresponding author name(s): Riccardo Di Clemente

Reviewer Comments & Decisions:

Decision Letter, initial version:

9th August 2022

Dear Dr Di Clemente,

Thank you once again for your manuscript, entitled "Changes in the time-space dimension of human mobility during the COVID-19 pandemic", and for your patience during the lengthier than usual peer review process.

Your Article has now been evaluated by 2 referees. You will see from their comments copied below that, although they find your work of potential interest, they have raised quite substantial concerns. In light of these comments, we cannot accept the manuscript for publication, but would be interested in considering a revised version if you are willing and able to fully address reviewer and editorial concerns.

We hope you will find the referees' comments useful as you decide how to proceed. If you wish to submit a substantially revised manuscript, please bear in mind that we will be reluctant to approach the referees again in the absence of major revisions. We are committed to providing a fair and constructive peer-review process. Do not hesitate to contact us if there are specific requests from the reviewers that you believe are technically impossible or unlikely to yield a meaningful outcome.

In your revision, we ask you to address all reviewers' concerns. Both reviewers believe that your work would benefit from an additional POI-based analysis, and we ask that you include this in your revision if possible given the structure of your dataset. Please contextualize your observations both in terms of prior literature on temporal mobility patterns, and the local social distancing policies throughout the pandemic. Finally, please discuss any biases and representativeness of your datasets.

Finally, your revised manuscript must comply fully with our editorial policies and formatting requirements. Failure to do so will result in your manuscript being returned to you, which will delay its consideration. To assist you in this process, I have attached a checklist that lists all of our requirements. If you have any questions about any of our policies or formatting, please don't hesitate

to contact me.

If you wish to submit a suitably revised manuscript we would hope to receive it within 4 months. I would be grateful if you could contact us as soon as possible if you foresee difficulties with meeting this target resubmission date.

- Include a "Response to the editors and reviewers" document detailing, point-by-point, how you addressed each editor and referee comment. If no action was taken to address a point, you must provide a compelling argument. When formatting this document, please respond to each reviewer comment individually, including the full text of the reviewer comment verbatim followed by your response to the individual point. This response will be used by the editors to evaluate your revision and sent back to the reviewers along with the revised manuscript.
- Highlight all changes made to your manuscript or provide us with a version that tracks changes.

[REDACTED]

Thank you for the opportunity to review your work. Please do not hesitate to contact me if you have any questions or would like to discuss the required revisions further.

Sincerely,

Arunas Radzvilavicius, PhD
Editor
Nature Human Behaviour

Reviewer expertise:

Reviewer #1: spatial data science, urban analytics, computational methods

Reviewer #2: has not submitted their review

Reviewer #3: spatial-temporal mobility analytics

REVIEWER COMMENTS:

Reviewer #1:
Remarks to the Author:

The article looks at the spatio-temporal changes in movement patterns during COVID-19 pandemic. Although several studies have tackled the issue of human mobility pattern change detection, joint examination of space-time in this context adds much novelty to the manuscript.

However, there are a few minor comments that need to be addressed:

Introduction

-CDR and LBS are known to be intrusive technologies to a certain extent and may raise privacy concerns. Are the participants' personal information preserved? A little detailed explanation on how this is done might be useful.

- The authors claim that the temporal analysis of mobility patterns w.r.t commuting patterns is missing from the COVID-19 literature might be an overstatement. There are several studies that do look at space-time jointly especially in the time geography context - please add sufficient literature to support your claims.

-Radius of gyration may be a useful spatial metric but home range has been traditionally used in movement pattern analysis by spatial ecologists - were any other spatial metrics considered for this study? Any reasons to choose just the radius of gyration (i.e availability of data, relevance to COVID-19 social distancing etc.) must be stated more clearly. The reasoning behind the choice of variables for both spatial and temporal analysis must be stressed upon with supporting literature.

-The term 'co-temporal occurrence of daily mobility motifs' to define the mobility synchronization metric is sort of confusing and needs to be stated in a more simplified form. What would it mean to policymakers in terms of examining mobility synchronization?

Results

-The authors state that the frequency components of the Fourier spectra have been examined as part of the timeseries analysis - are these generated from raw trips aggregated hourly or changes in trips? This needs to be clarified further. If the POIs were not considered in the study - how was proximity to certain locations accounted for in the study i.e home-work, home-shops etc.?

- The dips in mobility synchronisation - hypothesized by the authors as related to stay-at-home policies need to be supported with WHO guideline and local healthcare policies more broadly to contextualize the meaning of these changes.

- Are the trips used for the data only commute trips or are leisure trips also considered? Its not clearly stated. Trip purpose can generate different sets of results.

Discussion

Overall, the discussion section would benefit a lot from policy guidelines in the UK ontext that pinpoint to the immediate findings in the paper. Citing references to establish some of the hypothetical claims would strengthen the contributions of the manuscript.

Methods

- Are the trips unique or repetitions exist?
- How representative are the data? Is there an existing bias in the data based on sampling or use of smartphones - are urban and rural areas represented in an equivalent manner?
- Can POIs be included to further expand the spatial metric and make it more contextual?

Overall, the paper is a very interesting one although some loose ends need to be tied before it can be accepted. The figures are very clear and self-explanatory.

Reviewer #2:

None

Reviewer #3:

Remarks to the Author:

The goal of this paper, "Changes in the time-space dimension of human mobility during the COVID-19 pandemic", in review for "Nature Human Behavior", is to identify the nature of human mobility patterns in response to the COVID pandemic and lockdown policies imposed by local governments in the UK, with special attention paid to disaggregating mobility into its spatial and temporal dimensions. Furthermore, this paper measured how mobility changed for different socio-economic and urbanized locations around the UK. Many of the previously published mobility-focused COVID research focuses on measuring change in overall decreases in mobility, particularly measured by means of distance traveled, through time as it relates to lockdown restrictions. The efforts of this paper to disaggregate human mobility into distinct spatial and temporal dimensions is highly relevant not only for COVID-based research, but for broader human mobility work in general. The paper is generally well-written and sets up the problem statement and relevance of the work in a clear way. The approach and quality of the data are valid, and the results provide an appropriate answer that fills a gap in the research community. My comments below are aimed at improving the current version of the manuscript.

An important value this paper adds is to illustrate the detailed way in which the nature of mobility changed – to a deeper level than the previously established "drop in mobility" during COVID – particularly in the temporal dimension with their Mobility Synchronization metric. This disaggregation into spatial and temporal dimensions extends the current research to tell a more complete story in how human behavior changed during the pandemic. Furthermore, the approach to understanding temporal aspects of mobility by way of their Mobility Synchronization metric is valid and contributes to a larger research area of temporal dynamics of mobility, past the direct application for COVID. This is important and relevant work that contributes to broader research in human mobility.

While this paper correctly identifies a gap in temporal dimensions in mobility research, there are recent studies that have approached temporal measures of human activity, with some specifically in the context of COVID (Aledavood, 2022; Sparks, 2022; Leng, 2021). While the authors still present a novel approach to measuring the temporal dimension of human mobility, the authors should consider including these references to better present the state of temporal dynamics of mobility research.

The paper could benefit from more descriptive statistics about the original dataset. I would have preferred more clarity on the data to help illustrate the underlying data structure. For example, more detail on what one of the individual records of the 17.8 billion out-of-home trips look like would help clarify the underlying structure of the data. It is difficult to assess whether one of the records contains

data on the number of unique visitation locations out-of-home, the length stayed at a single location versus length of time spent away-from-home, etc. While some of this is illustrated in Table S1.1-2, the data structure is not clear, and further clarification would be appreciated. For example, the "trips" variable detailed in table S1.2 describes the total number of times users left home. The difference between a user leaving home, going to a single location and returning home, is different than a user leaving home, and going to multiple locations and returning home, even if they are both over the same unit of time, particularly in the context of covid exposure and general human mobility research. Further explanation on this distinction is needed.

While the authors show correlations with users in the dataset with census population numbers (Fig. S1.1), an r-squared value of 0.78 still represents arguably enough bias in the data to warrant brief commentary on who may or may not be represented in the data. I was left wondering about potential biases, such as the possible influence of super-users over-weighting the distribution of mobility activity. Were there any outliers in the data that were removed? Is there a possible influence of people moving during this time period and having their at-home location change more than once? If so, how might that influence Radius of Gyration?

Generally, mobility datasets may be described as "place-based" data products that present change in activity over time in bucketed places as a spatial unit of measure, and "individual-based" data products that present trajectories of individual agents over time. While I understand the data used in this paper seems to be a more "individual-based" perspective of mobility, and because of this, the authors could not add a perspective of visitations to points of interest (POIs), a bit more commentary on the possible nature of travel would benefit the paper. If the context of disaggregating mobility into spatial and temporal dimensions is for the purposes of covid, knowing whether people are taking long drives to public parks, or whether they're going to POIs where high probability of mixing events can occur greatly influences the spread of covid. This is not to say that an analysis is required, simply commentary on the influence of how the nature of travel might affect the perceived response to lockdowns would benefit the paper in the context of effectiveness of covid lockdown policies.

Smaller point, in the fourth paragraph of the introduction, "have" should be replaced by "has" in the following sentence, "...how each dimension have been reshaped during the pandemic."

References:

Aledavood, Talayeh, et al. "Quantifying daily rhythms with non-negative matrix factorization applied to mobile phone data." *Scientific reports* 12.1 (2022): 1-10.

Sparks, Kevin, et al. "Shifting temporal dynamics of human mobility in the United States." *Journal of Transport Geography* 99 (2022): 103295.

Leng, Yan, Dominiquo Santistevan, and Alex Pentland. "Understanding collective regularity in human mobility as a familiar stranger phenomenon." *Scientific Reports* 11.1 (2021): 1-9

Author Rebuttal to Initial comments

Reply to reviewer #1

1. *The article looks at the spatio-temporal changes in movement patterns during COVID-19*

pandemic. Although several studies have tackled the issue of human mobility pattern change detection, joint examination of space-time in this context adds much novelty to the manuscript.

Our reply: We would like to thank the reviewer for their careful review of the paper. We greatly appreciate their constructive comments to our manuscript. We also appreciate their interest and positive review of our manuscript. We apologise for the delayed submission driven by our efforts to collect more data to extend our analysis and further strength our contribution. Each comment was considered as detailed below. The reviewer can check all the changes within the manuscript in the changes file attached to the end of the submission. (in blue is the new text, and in red is the deleted one).

- 2. CDR and LBS are known to be intrusive technologies to a certain extent and may raise privacy concerns. Are the participants' personal information preserved? A little detailed explanation on how this is done might be useful.*

Our reply: We thank the reviewer for this comment. We understand that CDR and LBS can be very intrusive, but we want to emphasise that respecting the users' privacy is a must in our work. All the data is collected using a GDPR-compliant framework in which the users are aware and agree to have their data collected. From all the available data, we are only interested in mobility-related data, and we do not use personal or other information that would allow the identification of users. As can be seen from the description of the datasets built for this study (Section S1 in the Supplementary Information), all data is anonymised and aggregated to the level of the UK's local authorities.

Furthermore, our study does not use the user's specific locations. For the radius of gyration, we estimate the home location using a geofencing approach with a 200m radius of the most visited areas in a certain period. For the same reason, except for visits to green areas such as public parks, we do not analyse users' locations for POI identification. Even in the case of the recently added analysis of the green areas fruition, our dataset does not contains cemeteries and religious grounds from the data. All this has been described in Data Sources Section in the main manuscript and Section S1 in the Supplementary information.

- 3. The authors claim that the temporal analysis of mobility patterns w.r.t commuting patterns is missing from the COVID-19 literature might be an overstatement. There are several studies that do look at space-time jointly, especially in the time geography context - please add sufficient literature to support your claims.*

Our reply: We thank the reviewer for this comment. We took the reviewer's comment seriously and toned down our statement about the literature gap mentioning that our work adds to the literature on spatio-temporal studies of human mobility. Additionally, we revisited the literature, adding references in the introduction that composed the state-of-the-art and supported our arguments (references [59-61, 64-71, 72-77]).

- 4. Radius of gyration may be a useful spatial metric but home range has been traditionally used in movement pattern analysis by spatial ecologists - were any other spatial metrics considered for this study? Any reasons to choose just the radius of gyration (i.e availability of data, relevance to COVID-19 social distancing etc.) must be stated more clearly. The reasoning behind the choice of variables for both spatial and temporal analysis must be stressed upon with supporting literature.*

Our reply: We agree with the reviewer about other metrics, however the use of radius of gyration is quite standard in the literature of human mobility and the choice is hence based on this fact alone. By using the radius of gyration we ensure this work aligns with the literature better and hence become a source of comparison with other works on mobility. This explanation is included in the introduction, as well as references to articles [44, 47, 64-75] that support our claims.

It is important however acknowledge and discuss other, less standard metrics/variables such as the mobility synchronisation. We have added in the Introduction some explanation and motivation about mobility synchronisation indicating that human mobility patterns, especially regarding commuting behaviour, exhibit a high degree of temporal regularity. Our mobility synchronisation metric aims to measure these regularities, determined mainly by the synchronisation of labour, work schedules, and activities we accommodate around them.

- 5. The term 'co-temporal occurrence of daily mobility motifs' to define the mobility synchronization metric is sort of confusing and needs to be stated in a more simplified form. What would it mean to policymakers in terms of examining mobility synchronization?*

Our reply: We apologise for not providing a clear definition for the metric. We provided additional information to explain that mobility synchronisation aims to capture synchronised movements, such as leaving home at a specific time. In this context of a pandemic, high levels of synchronised movements might increase the virus transmission risk since more people are more likely to be in shared spaces simultaneously (e.g. public transport and offices). Hence, for policymakers, monitoring the levels of mobility synchronisation could be beneficial for designing and assessing the effectiveness of mobility restriction guidelines.

The modifications are highlighted in blue in the first paragraph of the Introduction, the last paragraph of the Discussion and the Supplementary Information Section S3.

- 6. The authors state that the frequency components of the Fourier spectra have been examined as part of the timeseries analysis - are these generated from raw trips aggregated hourly or changes in trips? This needs to be clarified further. If the POIs were not considered in the study - how was proximity to certain locations accounted for in the study i.e home-work, home-shops etc.?*

Our reply: We thank the reviewer for this relevant comment. We use the raw trip data aggregated at the hour level for the time-series analyses. In fact, for this analysis, we are interested only in the out-of-home component of the trip (i.e. the number of users who left home at each hour), so we do not need to identify the end location of the trips. However, in this revised version, we included the analysis of visits to green spaces, such as parks. The results of this can be seen in Figure 5. For this new analysis, we used a dataset of locations of this kind of POIs in England and, using geofencing, we identify when the user's trip intersects with green areas.

The altered text with the trips' explanation is in the main manuscript's first paragraph of the Results Section. The description of the data for green POIs in England and the new dataset created with the number of trips passing through them are given in Section S1 of the Supplementary Information and Section Other Data Sources in the main manuscript.

- 7. The dips in mobility synchronisation - hypothesized by the authors as related to stay-at-home policies need to be supported with WHO guideline and local healthcare policies more broadly to contextualize the meaning of these changes.*

Our reply: We thank the reviewer for this comment. We agreed with the reviewer and revisited the literature, adding references in the results and discussion sections that complemented and supported our hypotheses/findings (references [80-86]).

- 8. Are the trips used for the data only commute trips or are leisure trips also considered? Its not clearly stated. Trip purpose can generate different sets of results.*

Our reply: The authors thank the reviewer for raising this issue. All types of trips are considered in this analysis, and we considered that, during lockdown periods, the number of holidays trips would be negligible. In addition to that, to further respect users' privacy, we did not include analyses that require the identification of the destination of the trips. We focus only on the duration, time and frequency of these trips. We believe that providing valuable analysis of people's mobility while preserving their privacy is a strong point of our work.

Nonetheless, following Reviewer 3's suggestion, we decided to extend our analysis to cover trips to parks and other green areas. We changed the text to clarify the trip definition (fifth paragraph of the Introduction in the main manuscript and Section S3 in the Supplementary Information), and the new results can be seen in Figure 5. The results indicate changes in the duration and frequency of trips across different income (C) and urbanisation levels (D). Note that compared to the baseline year of 2019, we observed changes in all scenarios studied but in different magnitudes. For the work-related trips, we can see the higher managerial occupations presented the most degree of perturbation into their trip duration.

- 9. Overall, the discussion section would benefit a lot from policy guidelines in the UK context that*

pinpoint to the immediate findings in the paper. Citing references to establish some of the hypothetical claims would strengthen the contributions of the manuscript.

Our reply: We thank the reviewer for this comment. We agreed with the reviewer and added references [80-86] in the discussion that complemented and supported our findings and arguments.

10. Are the trips unique or repetitions exist?

Our reply: The authors thank the reviewer for raising this issue. Due to the method the data was collected, there can be repetition in the trips depending on the period considered. For example, within a week, multiple work trips from the same user can be recorded.

We adjusted the text to clarify the trip definition (fifth paragraph of the Introduction in the main manuscript and Section S3 in the Supplementary Information).

11. How representative are the data? Is there an existing bias in the data based on sampling or use of smartphones - are urban and rural areas represented in an equivalent manner?

Our reply: We thank the reviewer for this suggestion. We agree that additional representativeness and bias investigations are required, and we included new results in Section S1 of the Supplementary Information. There was no indication that a region or group could influence the result by overweighing the distributions. It is worth mentioning that, for the level of urbanisation and socioeconomic studies, we only use data from England due to the lack of a standard representation for all UK's countries. Moreover, before the analysis, we filter the data set and remove users with abnormal activity, such as too few logs in the server, lower location accuracy or a large range of motion. Furthermore, the home location was re-estimated every two weeks, so if a user moved to a different location, their home location would be updated for the calculations in the subsequent time interval. All this information was added to Section S1 of the Supplementary Information.

12. Can POIs be included to further expand the spatial metric and make it more contextual?

Our reply: We thank the reviewer for this suggestion. We agree that the study would benefit from the POIs expansion but we believe that due to the nature and aim of our study, an extensive study of POIs would be out of its scope. However, to provide some analysis in this area, we added the analysis of trips intersecting green areas such as parks, sports facilities and play areas. Besides giving us an general sense of leisure trips, this analysis would still preserve the user's privacy. Privacy issues is another reason why we do not include other types of POI. The new results in Figure 5 indicate a reduction in visits to these places during the announcement of the first lockdown in both urban and rural locations. However, after the end of the first lockdown, rural areas displayed a higher increase in visits when compared to urban.

The modifications and new results are illustrated in Figure 5 and the main manuscript's last two paragraphs of the Results Section. Additional information on the data, methods and other results of this analysis can be found in the supplemental information.

Reply to reviewer #3

- 1. The goal of this paper, "Changes in the time-space dimension of human mobility during the COVID-19 pandemic", in review for "Nature Human Behavior", is to identify the nature of human mobility patterns in response to the covid pandemic and lockdown policies imposed by local governments in the UK, with special attention paid to disaggregating mobility into its spatial and temporal dimensions. Furthermore, this paper measured how mobility changed for different socio-economic and urbanized locations around the UK. Many of the previously published mobility focused covid research focuses on measuring change in overall decreases in mobility, particularly measured by means of distance traveled, through time as it relates to lockdown restrictions. The efforts of this paper to disaggregate human mobility into distinct spatial and temporal dimensions is highly relevant not only for covid based research, but for broader human mobility work in general. The paper is generally well written and sets up the problem statement and relevance of the work in a clear way. The approach and quality of the data are valid, and the results provide an appropriate answer that fills a gap in the research community. My comments below are aimed at improving the current version of the manuscript.*

Our reply: We thank the reviewer for their thorough review and their constructive comments on our manuscript. Also, we are grateful for their interest in our manuscript and for the positive review. We apologise for the delayed submission driven by our efforts to collect more data to extend our analysis and further strength our contribution. Each comment was considered as detailed and all the changes within the manuscript in the changes file are highlighted (in blue is the new text, and in red is the deleted one).

- 2. An important value this paper adds is to illustrate the detailed way in which the nature of mobility changed to a deeper level than the previously established "drop in mobility" during covid particularly in the temporal dimension with their Mobility Synchronization metric. This disaggregation into spatial and temporal dimensions extends the current research to tell a more complete story in how human behavior changed during the pandemic. Furthermore, the approach to understanding temporal aspects of mobility by way of their Mobility Synchronization metric is valid and contributes to a larger research area of temporal dynamics of mobility, past the direct application for covid. This is important and relevant work that contributes to broader research in human mobility.*

Our reply: The authors thank the reviewer for recognising the value of our contribution to the

literature on the changes in human mobility during the pandemic. We hope that this new version further improves the quality of our work.

- 3. While this paper correctly identifies a gap in temporal dimensions in mobility research, there are recent studies that have approached temporal measures of human activity, with some specifically in the context of covid (Aledavood, 2022; Sparks, 2022, Leng, 2021). While the authors still present a novel approach to measuring the temporal dimension of human mobility, the authors should consider including these references to better present the state of temporal dynamics of mobility research.*

Our reply: We thank the reviewer for this comment. We agreed that the suggested references are relevant and should be added (references [59-61]) to our state-of-the-art description, and they were added to the third paragraph of the introduction in the main manuscript. We also revisited the literature and added references that complemented and supported our findings (references [64-71, 72-77]).

- 4. The paper could benefit from more descriptive statistics about the original dataset. I would have preferred more clarity on the data to help illustrate the underlying data structure. For example, more detail on what one of the individual records of the 17.8 billion out-of-home trips look like would help clarify the underlying structure of the data. It is difficult to assess whether one of the records contains data on the number of unique visitation locations out-of-home, the length stayed at a single location versus length of time spent away-from-home, etc. While some of this is illustrated in Table S1.1-2, the data structure is not clear, and further clarification would be appreciated. For example, the "trips" variable detailed in table S1.2 describes the total number of times users left home. The difference between a user leaving home, going to a single location and returning home, is different than a user leaving home, and going to multiple locations and returning home, even if they are both over the same unit of time, particularly in the context of covid exposure and general human mobility research. Further explanation on this distinction is needed.*

Our reply: The authors thank the reviewer for raising this issue. Due to privacy concerns, we do not use the exact location of user's homes. We estimate the home location using a geofencing approach with a 200m radius of the most visited areas in a certain period. Work places and other POI's, except public green areas, are not detected as well. We adjusted the text to clarify the term trip refers to the amount of time or distance travelled when the user leaves their home geo-fencing area and enters it again. This information was added to the fifth paragraph of the Introduction in the main text.

- 5. While the authors show correlations with users in the dataset with census population numbers (Fig. S1.1), an r -squared value of 0.78 still represents arguably enough bias in the data to warrant brief commentary on who may or may not be represented in the data. I was left wondering about potential biases, such as the possible influence of super-users over-weighting the distribution of mobility activity. Were there any outliers in the data that were removed? Is there a possible influence of people moving during this time period and having their at-home*

location change more than once? If so, how might that influence Radius of Gyration?

Our reply: We thank the reviewer for this suggestion. We agree that additional representativeness and bias investigations are required, and we analysed the data distribution by country and urbanisation level. There was no indication that a region or group could influence the result by over-weighting the distributions. Note that, for the level of urbanisation and socioeconomic studies, we only use data from England due to the lack of a standard representation for all UK's countries. Moreover, before the analysis, we filter the data set and remove users with abnormal activity, such as too few logs in the server, lower location accuracy or a large range of motion (e.g. users travelling more than 100km in a day). Furthermore, the home location was re-estimated every two weeks, so if a user moved to a different location, their home location would be updated for the calculations in the subsequent time interval. All this information was added to Section S1 of the Supplementary Information.

6. *Generally, mobility datasets may be described as "place-based" data products that present change in activity over time in bucketed places as a spatial unit of measure, and "individual-based" data products that present trajectories of individual agents over time. While I understand the data used in this paper seems to be a more "individual-based" perspective of mobility, and because of this, the authors could not add a perspective of visitations to points of interest (POIs), a bit more commentary on the possible nature of travel would benefit the paper. If the context of disaggregating mobility into spatial and temporal dimensions is for the purposes of covid, knowing whether people are taking long drives to public parks, or whether they're going to POIs where high probability of mixing events can occur greatly influences the spread of covid. This is not to say that an analysis is required, simply commentary on the influence of how the nature of travel might affect the perceived response to lockdowns would benefit the paper in the context of effectiveness of covid lockdown policies.*

Our reply: We agree and thank the reviewer for this comment. As the reviewer said, due to the nature and aim of our study, an extensive study of POIs would be out of its scope. However, we included comments in the discussion section about the nature of trips. We also collected a new dataset of trips intersecting green areas (e.g. green areas such as parks, sports facilities and play areas). The new results indicate a reduction in visits to these places during the announcement of the first lockdown in both urban and rural locations. However, after the end of the first lockdown, rural areas displayed a higher increase in visits when compared to urban. We also present results related to the length of trips starting when people usually leave for work. Although we can not state that all those trips are work-related, this approach can capture the overall trends in the work trips.

The modifications and new results are illustrated in Figure 5 and the main manuscript's last two paragraphs of the Results Section. Additional information on the data, methods and other results of this analysis can be found in the supplemental information.

7. *Smaller point, in the fourth paragraph of the introduction, "have" should be replaced by "has" in the following sentence, "how each dimension have been reshaped during the pandemic."*

Our reply: The authors thank the reviewer for raising this issue. We highlight that we revised the entire manuscript to fix typos and other grammar issues.

Decision Letter, first revision:

13th April 2023

Dear Dr. Di Clemente,

Thank you for submitting your revised manuscript "Changes in the time-space dimension of human mobility during the COVID-19 pandemic" (NATHUMBEHAV-22051134A). It has now been seen by the original referees and their comments are below. As you can see, the reviewers find that the paper has improved in revision. We will therefore be happy in principle to publish it in Nature Human Behaviour, pending minor revisions to satisfy the referees' final requests and to comply with our editorial and formatting guidelines.

We are now performing detailed checks on your paper and will send you a checklist detailing our editorial and formatting requirements within a week. Please do not upload the final materials and make any revisions until you receive this additional information from us.

Sincerely,

Arunas Radzvilavicius, PhD
Senior Editor, Nature Human Behaviour
Nature Research

Reviewer #1 (Remarks to the Author):

The revised manuscript seems up to the mark with minor grammatical and sentence construction errors. It is OK to accept for publication.

Reviewer #3 (Remarks to the Author):

The authors addressed all my comments, suggestions, and concerns to an acceptable degree. Their response was thorough and improved the quality of the manuscript. It is interesting research that contributes to the science of human mobility. I hope my comments were helpful and look forward to reading the final published manuscript.

Author Rebuttal, first revision:

Reply to reviewer #1

1. The revised manuscript seems up to the mark with minor grammatical and sentence construction errors. It is OK to accept for publication.

Our Reply: We want to thank the reviewer for their careful review of the paper and valuable comments, which improved the quality of our contribution. We greatly appreciate their constructive comments on our manuscript. We revised the text and corrected the grammar and other issues to comply with the journal's guidelines.

2. The authors addressed all my comments, suggestions, and concerns to an acceptable degree. Their response was thorough and improved the quality of the manuscript. It is interesting research that contributes to the science of human mobility. I hope my comments were helpful and look forward to reading the final published manuscript.

Our Reply: We thank the reviewer for their thorough review and constructive comments on our manuscript. Also, we are grateful for their interest in our manuscript, the positive review, and for considering it worth publication in *Nature Human Behaviour*. In this new revision, we fixed minor grammar issues and made the paper fully compliant with the journal's guidelines.

Final Decision Letter:

Dear Professor Di Clemente,

We are pleased to inform you that your Article "COVID-19 is linked to changes in the time-space dimension of human mobility", has now been accepted for publication in *Nature Human Behaviour*.

Please note that *Nature Human Behaviour* is a Transformative Journal (TJ). Authors whose manuscript

was submitted on or after January 1st, 2021, may publish their research with us through the traditional subscription access route or make their paper immediately open access through payment of an article-processing charge (APC). Authors will not be required to make a final decision about access to their article until it has been accepted. IMPORTANT NOTE: Articles submitted before January 1st, 2021, are not eligible for Open Access publication. Find out more about Transformative Journals

With best regards,

Arunas Radzvilavicius, PhD
Senior Editor, Nature Human Behaviour
Nature Research

P.S. Click on the following link if you would like to recommend Nature Human Behaviour to your librarian
<http://www.nature.com/subscriptions/recommend.html>

** Visit the Springer Nature Editorial and Publishing website at www.springernature.com/editorial-and-publishing-jobs for more information about our career opportunities. If you have any questions please click here.**

This email has been sent through the Springer Nature Tracking System NY-610A-NPG&MTS

Confidentiality Statement:

This e-mail is confidential and subject to copyright. Any unauthorised use or disclosure of its contents is prohibited. If you have received this email in error please notify our Manuscript Tracking System Helpdesk team at <http://platformsupport.nature.com>.

Details of the confidentiality and pre-publicity policy may be found here <http://www.nature.com/authors/policies/confidentiality.html>